# Nonlinear mechanics of hybrid polymer networks that mimic the complex mechanical environment of cells

Maarten Jaspers[1], Sarah L. Vaessen[1], Pim van Schayik[1], Dion Voerman[1], Alan E. Rowan[1,2] & Paul H.J. Kouwer[1]

The mechanical properties of cells and the extracellular environment they reside in are governed by a complex interplay of biopolymers. These biopolymers, which possess a wide range of stiffnesses, self-assemble into fibrous composite networks such as the cytoskeleton and extracellular matrix. They interact with each other both physically and chemically to create a highly responsive and adaptive mechanical environment that stiffens when stressed or strained. Here we show that hybrid networks of a synthetic mimic of biological networks and either stiff, flexible and semi-flexible components, even very low concentrations of these added components, strongly affect the network stiffness and/or its strain-responsive character. The stiffness (persistence length) of the second network, its concentration and the interaction between the components are all parameters that can be used to tune the mechanics of the hybrids. The equivalence of these hybrids with biological composites is striking.

[1] Department of Molecular Materials, Radboud University Nijmegen, Institute for Molecules and Materials, Heyendaalseweg 135, 6525 AJ Nijmegen, The Netherlands. [2] The University of Queensland, Australian Institute for Bioengineering and Nanotechnology, Brisbane, Queensland 4072, Australia. Correspondence and requests for materials should be addressed to A.E.R. (email: alan.rowan@uq.edu.au) or to P.H.J.K. (email: p.kouwer@science.ru.nl).

ife is supported by fibrous polymer networks. The biopolymers that form these networks are found both inside cells (the cytoskeleton) and in the environment cells reside in (the extracellular matrix, ECM)[1]. The mechanical properties of the networks play a crucial role in all essential cellular processes, including differentiation, proliferation, transportation and communication[2,3]. A broad spectrum of forces are involved in these cellular functions, which require materials with different mechanical properties that are simultaneously active and responsive. Indeed, *in vivo* biopolymer networks composed of very soft (for example, glycans) to very stiff (for example, microtubules) components exist side by side. Physical and or chemical interactions between the network components in these hydrogels create a complex and amazingly adaptive mechanical environment. In recent years, numerous synthetic and semi-synthetic gels have been developed for biomedical applications[4,5], often with the goal to mimic Nature's intricate mechanics with synthetic materials[6], but even responsive 'smart' hydrogels have proved to be inadequate.

To more closely mimic these biological networks, composite or hybrid hydrogels of multiple polymer networks, either conjugated or only mechanically interlocked, have been developed. Composites of two synthetic flexible polymers show tremendous mechanical strength and toughness compared to their single-component equivalents[7–9] and composites with clay or metal-oxide nanosheets exhibited enhanced mechanical strength[10,11], as well as additional features such as self-healing properties[12] and anisotropic mechanics[13]. Although mechanically well understood, these materials are unlikely candidates to serve as synthetic ECM mimics, since they are composed of flexible polymer chains and do not form the fibrous structures that are found for biopolymers.

The difference in architecture between flexible synthetic and biopolymer gels is accentuated under strain[14]. Reconstituted networks of cytoskeletal polymers such as actin or intermediate filaments or extracellular biopolymers such as collagen or fibrin become many times stiffer on deformation[15–18]. This strain-

stiffening response prevents the networks from breaking under external stresses and also enables direct communication between cells[19,20]. As such, biopolymer hydrogels are an obvious choice as artificial ECM materials. In fact, composites of biopolymers are increasingly applied[4], for instance, collagen-alginate composites[21,22] among many others, although surprisingly little is known on how this desired stiffening response is affected by the composite nature of natural biopolymer networks and the effect of interactions between the components[23–25].

Here we describe how the mechanics of fibrous networks change in the presence of a second network component. We use a synthetic biopolymer mimic based on polyisocyanopeptides[26] (PICs), which form hydrogels that closely resemble both the fibrillar structure and strain-stiffening response of biopolymer gels[27–29]. PIC composite hydrogels are constructed by combining this biopolymer mimic with; rigid fibres (carbon nanotubes), semi-flexible (fibrin) bundles and flexible (polyacrylamide) polymers. Characterization of the composites allows us to study the changes in the linear and nonlinear mechanical properties, which we relate to natural fibrous networks, where the different components of the intracellular and extracellular matrix display similar effects[1,30]. The results reveal how readily the gel mechanics can be manipulated, even at very low concentrations of either component. The synthetic composites provide a guide on how to more closely mimic the complex mechanical behaviour of natural biopolymer composites, but also on the design of much 'smarter' soft materials.

## Results

**Materials and methods.** The structure of the hydrogel composites is schematically shown in Fig. 1. The primary component of all gels is the (ethylene glycol)-functionalized PIC, a helical, semi-flexible synthetic polymer[26,31]. This biomimetic synthetic polymer offers the advantage of excellent molecular control and can additionally be equipped with virtually any desired functional

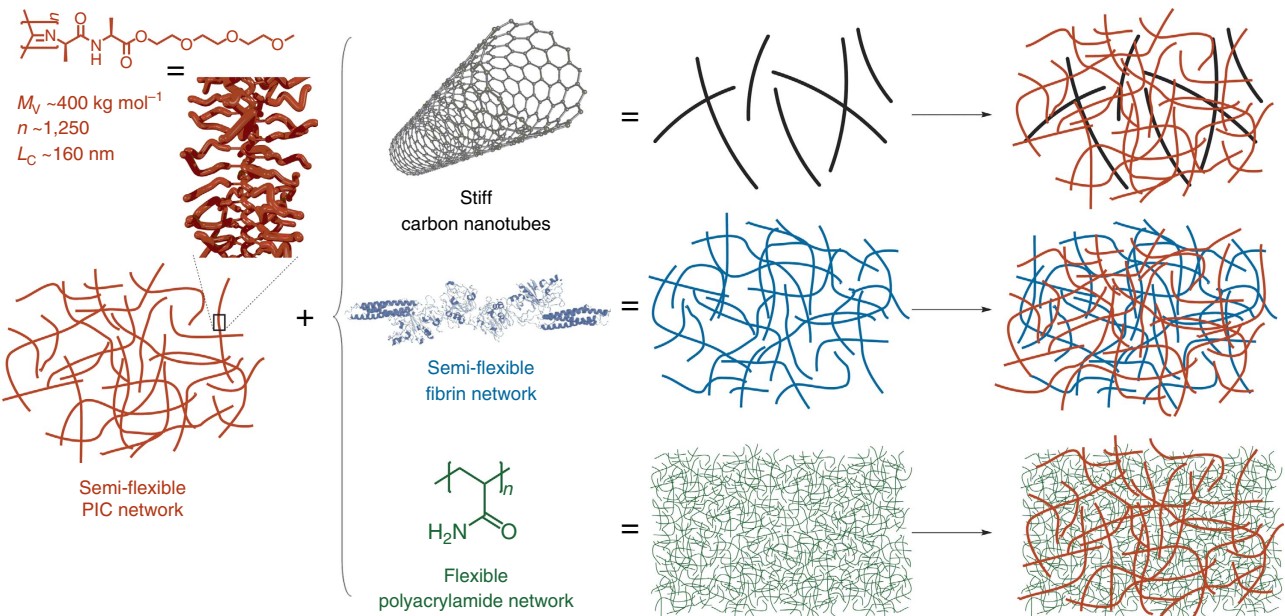

**Figure 1 | Composite hydrogel components.** The primary component of each hydrogel is a network of semi-flexible ethylene glycol-functionalized PICs (shown in red), which are long polymers trapped in a relatively stiff helical conformation. Heating an aqueous PIC solution above its gelation temperature induces bundling and gel formation. The second component of the hybrid hydrogels used are stiff CNTs (black), a network of semi-flexible fibrin fibres (blue) or a network of flexible PAAm polymers (green). The CNTs become interlocked in the PIC network on gelation, because their length is many times the PIC gel pore size. The other two hybrid gels are formed by polymerization of either fibrinogen or AAm monomers in the presence of a pre-formed PIC network. The double-network composites can be only mechanically interlocked (interpenetrating networks) or conjugated covalently.

In the figure labels:
$M_V$ ~400 kg mol$^{-1}$
$n$ ~1,250
$L_C$ ~160 nm

Stiff carbon nanotubes

Semi-flexible fibrin network

Semi-flexible PIC network

Flexible polyacrylamide network

group at its periphery, including peptides, dyes and proteins[32]. The ethylene glycol side chains grafted on the polymer backbone render aqueous PIC solutions thermoresponsive. At low temperatures, the polymer diffuses freely in solution. Heating the solution above the gelation temperature of 18 °C causes aggregation of the polymers into bundles with a persistence length $l_p$ of the order of hundreds of nanometres[26,29]. The hydrogel formed by the polymer bundles is fully elastic and shows many similarities with reconstituted gels of intermediate filaments or F-actin[26,27]. Recent stem-cell studies with peptide-modified PIC hydrogels underline how strongly gel composition and mechanics can affect differentiation outcomes in soft matrices[33].

For the semi-flexible/stiff composite hydrogels, we use single- and multi-walled carbon nanotubes (CNTs), which are composed of individual graphene sheets rolled up to make hollow structures. Both have high aspect ratios with diameters of molecular dimensions and lengths of microns and are among the stiffest synthetic filaments known with persistence lengths up to millimetres[34]. The combination of stiff filaments with a semi-flexible polymer network is highly relevant for the natural cytoskeleton, where a dilute network of similarly stiff microtubules exists within a denser network of semi-flexible actin and intermediate filaments[6].

The semi-flexible fibrin fibres used for the semi-flexible/semi-flexible composites are formed through polymerization of fibrinogen monomers, which is initiated by the enzyme thrombin. Typically, the fibres have persistence lengths of tens of microns and forms a network through crosslinking by factor XIII[35,36]. Its porous gel structure (mesh size of about 1–2 μm) is ideally suited as a three-dimensional cell culture matrix[37]. The combination of a fibrin network with PIC either mechanically intertwined or conjugated yields semi-flexible hybrids that have analogues in the cytoskeleton (actin and intermediate filaments) as well as in the extracellular matrix (collagen and fibrin).

For the last class of hybrid gels, we study the conventional (crosslinked) polyacrylamide (PAAm), which forms a dense network of flexible polymer chains. Although intensely studied as substrate material, its architecture renders PAAm gels less suitable as synthetic three-dimensional ECM mimic. Combining this model, flexible network with PICs yields both interlocked and covalently linked flexible/semi-flexible networks. These composites more closely mimic the extracellular matrix, where semi-flexible collagen and fibrin fibres coexist with more flexible protein fibres such as elastin and polysaccharides like hyaluronic acid[38].

For the PIC/fibrin and PIC/PAAm hybrids, the PIC gel is formed first and the second component polymerizes through the pores of the PIC network with a mesh size of about 100 nm. PIC/CNT hybrids are simply formed by heating a cold co-solution beyond the PIC gelation temperature. In none of the hybrids, we find any indication of (micro)phase separation. The CNT and PAAm hybrid gels are optically transparent and also in the PIC/fibrin system, gel formation is mutually unaffected[39]. In addition, preliminary small angle X-ray scattering experiments on the composite gels, simply show the sum of scattering patterns of the two components.

The mechanical properties of the composites and pure materials were studied with rheometry, after *in situ* preparation of the gels between the rheometer plates (details are given in the 'Methods' section). In a standard frequency sweep, that is, at low stress $\sigma$ (and low strain $\gamma$), we probe the linear viscoelastic regime, which is dominated by the storage modulus $G'$ for all PIC containing samples. Typically, values of the loss modulus $G''$ are at least an order of magnitude smaller at all frequencies.

At higher stresses, we enter the nonlinear mechanical regime. As is characteristic for fibrous networks, the modulus increases and becomes a function of the applied strain or stress. Rather than a standard strain sweep or LAOS experiment, we probe this regime with the pre-stress protocol, which gives the same results, but is a more gentle procedure[40]. We find that the resulting differential modulus $K' = \partial\sigma/\partial\gamma$ describes of the nonlinear regime more accurately than the shear modulus $G'(\gamma)$. Note that in the linear regime, at low stress, $K' = G'$. Supplementary Fig. 1 shows a scheme with frequently used different representations of $K'$ and $G'$ as well as an experimental comparison between a strain sweep and a pre-stress method on a PIC gel.

**Hybrid networks with stiff CNTs.** Three commercially available CNT samples were studied: multi-walled (length $L \approx 10$ μm, diameter $d \approx 20$ nm), single-walled ($L \approx 10$ μm, $d \approx 1.8$ nm) and short single-walled CNTs ($L \approx 1$ μm, $d \approx 1.8$ nm). Lengths and diameters were given by the supplier (see 'Methods' section); the latter were confirmed by electron microscopy experiments (Supplementary Fig. 2). The persistence length $l_p$ of multi-walled CNTs is in the millimetre range, while that of the more flexible single-walled CNTs is about 10–100 μm. For all tubes, $l_p$ is larger than their contour length, which in turn exceeds the mesh size $\xi$ of the PIC hydrogels ($\xi_{PIC} \approx 100$ nm)[26]: $l_{p,CNT} \gg L_{CNT} \gg \xi_{PIC}$. As such, the nanotubes become mechanically trapped in the porous structure of the gel and are limited in their movement. The commercial CNTs are obtained as stable concentrated aqueous solutions and were diluted before use. Typical CNT concentrations are below 0.1 wt-%, in line with the low microtubules concentrations in the cytoskeleton[41]. At the highest concentration that we use, the CNT solutions are in the semi-dilute regime, far below the Onsager transition. Although in this regime steric interactions may be present, our control experiments (Supplementary Fig. 3) did not show significant mechanical effects in the frequency and strain rate regimes that we study.

The aqueous dispersions of polymer-stabilized CNTs were mixed with cold PIC solutions and heated to 37 °C in the rheometer to form a hybrid hydrogel. Gel formation is not affected by the tubes or the stabilizer and the gelation temperature remains unaltered (Supplementary Fig. 4). In the linear viscoelastic regime, the mechanical properties of all PIC/CNT samples are dominated by the storage modulus $G'$ (Fig. 2a,b and Supplementary Fig. 5). Both $G'$ and $G''$ are virtually independent of frequency and are not impacted by the presence of the CNTs. The corresponding high-stress (at pre-stress $\sigma_0 = 40$ Pa) differential moduli $K'$ and $K''$ show a similar frequency (in)dependence.

After a critical stress is reached, the elastic modulus of the samples starts to increase and $K'$ becomes a function of applied $\sigma$ (Fig. 2c). The effects of CNT addition on the hydrogel mechanics are relatively small compared to varying the PIC concentration. They become clearer when $K'$ is plotted against the CNT concentration (Fig. 2c–g). Here we need to distinguish the linear and the nonlinear regime. In the linear regime, the CNTs (at either concentration $c_{PIC} = 1$ or 2.5 mg ml$^{-1}$) have no visible effect on the linear storage modulus $K' = G'$ (panels e,g). At these low concentrations, the CNTs are mechanically interlocked in the PIC network and do not contribute to the macroscopic stiffness of the hydrogel. In our experiments, we do not reach the high strain rates where CNT effects are anticipated. Analogous results are observed in composite hydrogels of polyethylene glycol and CNTs[42] and in reconstituted networks of F-actin with microtubules[24]. In examples where CNTs interact with a polymer host (gelatin methacrylate or DNA) to become an

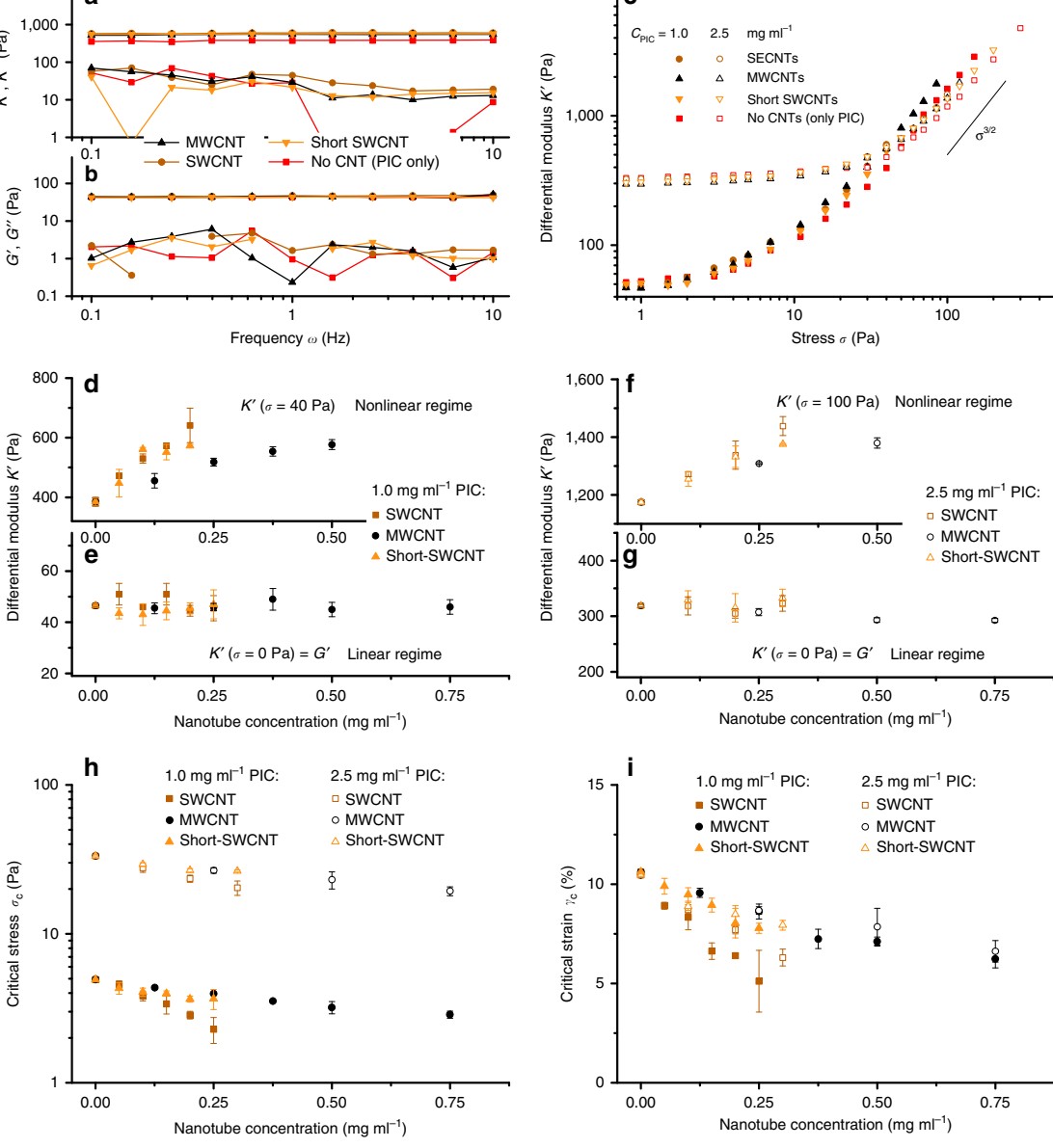

**Figure 2 | Mechanical properties of composites with semi-flexible PIC networks and rigid CNT rods.** (**a,b**) Frequency sweeps in the linear viscoelastic regime (**a**) and in the nonlinear regime at a pre-stress $\sigma_0 = 40$ Pa (**b**). The colours are indicated in the legend; the concentrations: $c_{PIC} = 1.0$ mg ml$^{-1}$ (all samples), $c_{SWCNT} = 0.2$ mg ml$^{-1}$, $c_{MWCNT} = 0.5$ mg ml$^{-1}$, $c_{short\text{-}SWCNT} = 0.2$ mg ml$^{-1}$. (**b**) Differential modulus at frequency $\omega = 1$ Hz at the same concentrations as in **a,b**. (**c–g**) Stiffness of PIC/CNT composites in the linear regime represented by the differential modulus $K'$ at $\sigma = 0$ Pa (**e,g**) and the nonlinear regime represented by $K'$ at $\sigma = 40$ Pa or $\sigma = 100$ Pa (**d,f**) for gels with PIC concentrations of 1.0 (**d,e**) and 2.5 mg ml$^{-1}$ (**f,g**), respectively. (**h**) The critical stress $\sigma_c$ decreases with nanotube concentration for gels with PIC concentrations of 1.0 and 2.5 mg ml$^{-1}$. Note the logarithmic scale on the y axis. (**i**) The critical strain $\gamma_c$ also decreases with increasing nanotube concentrations. The effects are strongest for the gels with long single-walled CNTs and low PIC concentrations. The error bars represent s.d.'s over two samples.

efficient crosslinker, however, the modulus increases with CNT content[43,44].

At higher stress in the nonlinear regime, $K'$ increases with increasing CNT concentrations (Fig. 2d,f). Already at $\sigma = 40$ Pa for the 1 mg ml$^{-1}$ PIC hydrogels and at $\sigma = 100$ Pa for the 2.5 mg ml$^{-1}$ PIC hydrogels, the rigid CNTs almost double the stiffness of the composites. The increased stiffening response of the gels originates from a reduced critical stress $\sigma_c$, which is the stress at which the nonlinear stiffening regime is entered and where the material becomes responsive to additional stress (Fig. 2h). This lower $\sigma_c$ means that composite gels with a higher CNT concentration are more sensitive to an applied stress, which

leads to an increased gel stiffness at stresses above $\sigma_c$ (Supplementary Fig. 6). For all three types of CNTs, $\sigma_c$ decreases with increasing nanotube concentration (Fig. 2h), but the effect is clearly smaller for multi-walled CNTs and for the shorter single-walled CNTs. For the 10 μm long single-walled CNTs, a concentration of as low as 0.005 wt% is already sufficient to lower $\sigma_c$ and increase the stiffening response of the PIC hydrogels.

For semi-flexible polymer networks, such as the PIC hydrogels, the low-stress linear regime is dominated by bending of the fibres, while the high-stress nonlinear regime is dominated by stretching the fibres[45,46]. The latter regime remains unaffected in the

composite, illustrated by the constant stiffening index $m = 3/2$ in $K' \propto \sigma^m$ observed at high stress (Supplementary Fig. 6)[15,47]. Instead, the CNTs shift the transition from the bending to the stretching dominated regime to lower stress, by suppressing heterogeneities in the deformation of the PIC network. Qualitatively, this effect was already observed in composites of F-actin and microtubules, where the stiff microtubules cause a more affine deformation of the surrounding actin network[24].

The effect becomes even clearer when we calculate the critical strain $\gamma_c = \sigma_c/G'$, which corresponds to the minimal deformation at which the material starts to stiffen up (Fig. 2i); $\gamma_c$ decreases by about a factor two at a $\sim 5\,nm$ ($0.25\,mg\,ml^{-1}$) concentration if single-walled CNTs in the hydrogel. The effect is less pronounced for the shorter CNTs, which emphasizes that longer CNTs can supress strain heterogeneities over much larger length scales. The observed difference between single- and multi-walled CNTs is simply the result of the much larger molecular weight of the latter. When one considers that the molar concentration of the multi-walled tubes ($0.25\,mg\,ml^{-1} \approx 50\,pM$) is about $100\times$ lower than that of the single-walled tubes, the effects of the much more rigid multi-walled CNTs are astounding. Figure 2i also shows that the CNTs have a slightly smaller effect on the gels with a higher PIC concentration, which is related to the already more affine deformation of a denser PIC network.

The results show that stiff rods, such as CNTs, are an effective tool to manipulate the mechanical properties of hydrogels, even at minimal concentrations. Without any specific interaction of the nanotubes with the network, no contribution to the gel modulus in the linear regime will be found. The main effect, however, is the enhanced sensitivity to stress that leads to an increased gel stiffness in the nonlinear regime. The concentration, length and persistence length of the rigid rods all contribute to the magnitude of this effect and we found the effect to be stronger at larger network mesh size $\xi$, provided that $L_{rod} \gg \xi$.

**Hybrid networks with a semi-flexible fibrin network.** Fibrin fibres are semi-flexible with persistence lengths of tens of microns and form porous gels after crosslinking[17]. Combining the synthetic PICs with a biopolymer, such as fibrin, not only results in hybrid hydrogels with tuneable material properties, but also leads to hydrogels, which are suitable for cellular studies. Previously, we showed that spreading and differentiation of human mesenchymal stem cells can be manipulated on hybrid hydrogels with different PIC to fibrin ratios[39]. Here we discuss the mechanical implication of combining two strain-stiffening networks of semi-flexible polymers[23]. Simulations on interpenetrating semi-flexible networks predict very rich mechanical properties, depending on the persistence length and ratio of the two components[25].

Fibrin gelation is initiated by thrombin and is clearly marked by the large increase in modulus after about 15 min (Fig. 3a, red dots). To form the composite hydrogel, a cold PIC solution in cell culture medium with thrombin was mixed with a cold fibrinogen solution in PBS buffer. Heating this combined solution to $T = 37\,°C$ results in fast PIC network formation and the much slower initiation of fibrin network formation, which again is marked by the increase in $G'$ after $\sim 15\,min$ (purple squares). At the chosen concentrations and conditions, the PIC and fibrin networks have similar stiffnesses. The linear modulus of the composite or double network equals the sum of moduli of the individual components. The PIC/fibrin hybrids show no significant relaxation at the time scale of the experiment (Supplementary Fig. 7).

The order of network formation may be important for the properties of the composite. Through the slow polymerization process of fibrinogen, fibrin grows in the already established PIC network. By using the thermoresponsive nature of the PIC hydrogels, the order can also be reversed (Fig. 3b). When the composite is cooled below the PIC gelation temperature, the PIC network selectively disassembles and the polymers dissolve, which leaves only the fibrin network with a modulus similar to the sample prepared in the absence of PIC. Reheating the sample causes a sharp increase in $G'$ at the PIC gelation temperature and at $T = 37\,°C$, the modulus is fully restored indicating that the two networks are mutually compatible and are not hindered by the presence of the other.

Although both PIC and fibrin individually form strain-stiffening hydrogels, their stiffening response to stress is very different. Whereas PIC hydrogels show the typical $K' \propto \sigma^{3/2}$ response resulting from stretching the semi-flexible fibres, fibrin hydrogels in the same stress regime show a much weaker stiffening response of about $K' \propto \sigma^{0.75}$ (Supplementary Fig. 7). This weaker response results from pulling out network fluctuations, and it is at only much higher stresses that the single fibres are stretched and the fibrin gels show the expected stiffening index $m = 3/2$ (ref. 17). The PIC/fibrin hybrid hydrogels exhibit a stiffening response that depends on the ratio of the two components (Fig. 3c). A composite hydrogel with $1\,mg\,ml^{-1}$ of both components (Table 1, entry 2) resembles the mechanical properties of the pure PIC hydrogel with approximately $K' \propto \sigma^{3/2}$, but when the fibrin concentration is increased to $5\,mg\,ml^{-1}$ (entry 6) the stiffening response is similar to pure fibrin hydrogels with $K' \propto \sigma^{0.75}$. By modifying the network composition, we can simply tailor the nonlinear mechanical response of the composite material. The intermediate stiffening response of the composites indicates a regime where both networks contribute to the nonlinear mechanics of the composite material. On applying a stress, both networks are deformed independently and in the absence of any specific interactions between them, they are only restricted by their mutual interpenetrated structure. In line with the predictions from simulations[25], we find that the mechanics are not simply a linear combination of the two components, but show a much richer nonlinear behaviour.

Assuming that the PIC and fibrin networks indeed strain stiffen independently in the composite hydrogels, their mechanics are expected to change by introducing specific binding interactions between the two networks. To study this, we functionalized[32,33] polyisocyanides with AKQAGDV peptides (0.2% functionalization, see Supplementary Methods and Supplementary Fig. 8 for details), which is the C-terminal end of the $\gamma$ chain of fibrinogen and is involved in the crosslinking of fibrin networks by factor XIII through the formation of $\gamma$ dimers[48,49]. The functionalized PIC hydrogels are a little stiffer than the regular PIC hydrogels (Table 1, entry 1), probably due to a slight increase in PIC chain length and a change in polymer hydrophilicity because of the attached peptides. Conjugated composite hydrogels at different PIC to fibrin ratios were prepared analogously to the aforementioned interpenetrating networks.

The mechanical properties of the conjugated hybrid hydrogels are quite different from the hydrogels with non-functionalized PICs, both in the linear and nonlinear regime (Fig. 3d and Table 1). At low stress, we observe an increase in stiffness when adding only a small amount of fibrin (Table 1, entry 2). Remarkably, this increase in stiffness is already observed in the PIC network, that is, before the fibrin network has formed (Supplementary Fig. 9). We can attribute this to the binding of fibrinogen molecules to the AKQAGDV peptides attached to the PIC network, which effectively results in the formation of additional crosslinkers in the PIC network causing the increased stiffness. The nonlinear mechanics of the conjugated hybrid hydrogels show for all compositions an

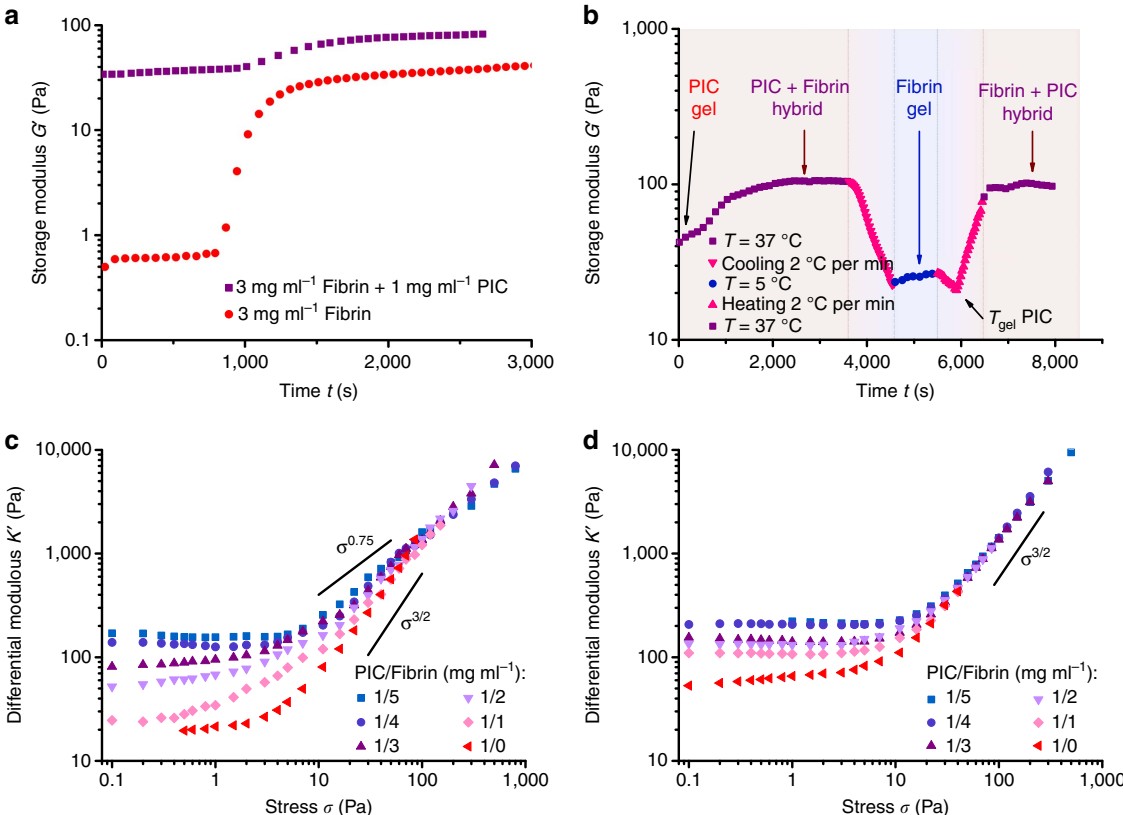

**Figure 3 | Mechanical properties of composites with semi-flexible PIC networks and semi-flexible fibrin networks.** (**a**) Time trace of the storage modulus $G'$, showing the formation of the fibrin network in the presence of the PIC network (black squares) and without the PIC network (red dots) at $T = 37\,°C$. In the composite, the starting modulus at $t = 0\,s$ is much higher, because the PIC hydrogel is formed already. (**b**) Time trace of $G'$ of the composite PIC (1 mg ml$^{-1}$)/fibrin (3 mg ml$^{-1}$) hydrogel through a cooling and heating cycle. At $t = 0$, only the PIC network is present; at $t \approx 3,000\,s$, $G'$ has PIC and fibrin contributions; cooling shows $G'$ of fibrin only and reheating restores the modulus completely. (**c**) Differential modulus $K'$ as a function of stress $\sigma$ for PIC/fibrin interpenetrating networks with varying PIC to fibrin mass ratios, showing different stiffening responses. The solid lines represent the stiffening responses $K' \propto \sigma^m$ with $m = 0.75$ for fibrin and $m = 3/2$ for PIC. (**d**) Conjugated PIC/fibrin hybrid hydrogels with AKQAGDV peptide-functionalized PICs (0.2%) all show a similar stiffening response, with $K' \propto \sigma^{3/2}$.

**Table 1 | Mechanical properties of conjugated and unconjugated PIC/fibrin hybrid hydrogels at different component ratios.**

| Entry | Gel composition | | Interpenetrating networks with unfunctionalized PIC | | | Conjugated networks with peptide-functionalized PIC | | |
|---|---|---|---|---|---|---|---|---|
| | PIC (mg ml$^{-1}$) | Fibrin (mg ml$^{-1}$) | $G'$ (Pa) | $\sigma_c$ (Pa) | $m$ | $G'$ (Pa) | $\sigma_c$ (Pa) | $m$ |
| 1 | 1 | 0 | 20 | 2.9 | 1.5 | 46 | 4.7 | 1.5 |
| 2 | 1 | 1 | 25 | 1.7 | 1.2 | 110 | 9.5 | 1.4 |
| 3 | 1 | 2 | 52 | 3.4 | 1.0 | 130 | 11 | 1.3 |
| 4 | 1 | 3 | 83 | 4.4 | 0.95 | 160 | 15 | 1.3 |
| 5 | 1 | 4 | 140 | 6.7 | 0.89 | 210 | 20 | 1.4 |
| 6 | 1 | 5 | 180 | 7.7 | 0.79 | 230 | 21 | 1.3 |

identical stiffening response with approximately $K' \propto \sigma^{3/2}$ (Fig. 3d). The conjugation prevents independent deformation of the two networks and the high-stress mechanics are dominated by the component with the strongest stiffening response, which is the PIC network. Even at high fibrin content, the composite gels adhere to the nonlinear response of $K' \propto \sigma^{3/2}$.

In short, the mechanics of hybrid hydrogels of two semi-flexible networks depend strongly on the composition and the conjugation between the two components. Without any specific interactions, the linear modulus is merely a linear combination of the two components and the nonlinear behaviour intermediate of the two pure networks. On deformation, both networks are deformed independently and they both contribute to the nonlinear stiffening response of the composite hydrogel. When specific binding interactions between the networks are introduced, however, the linear modulus of the gel increases due to the formation of additional network crosslinks and the strain-stiffening response is dominated by one of the components, since the two networks cannot be deformed independently any further. Interestingly, in case of reversible conjugation—not studied in this manuscript—also binding time scales will play a prominent role and could be used to cleverly tune the hydrogel mechanics[50].

**Hybrid networks with a flexible PAAm network.** For the last class of hybrid hydrogels, we combine the PIC network with networks of flexible PAAm chains with a persistence length in the nanometre range. Such flexible polymer hydrogels do not show a strain-stiffening response, but their stiffness, controlled by the monomer and crosslinker concentration, simply remains constant up to large stresses or deformations[14].

To form a PIC/PAAm composite hydrogel, a cold PIC solution was mixed with a solution of acrylamide (AAm) monomer and a small quantity of N,N'-methylenebisacrylamide (MBAA) crosslinker. The addition of a free radical initiator ($K_2S_2O_8$) and heating this solution to $T = 50\,°C$ starts the polymerization of the PAAm network. In this way, the PAAm network is formed within the PIC network in a similar way as for the PIC/fibrin hydrogels.

We study the mechanical properties of composites prepared from PIC (1 mg ml$^{-1}$) and AAm (21–57 mg ml$^{-1}$ or 0.3–0.8 M) with fixed amounts of MBAA crosslinker (0.5 mol% of AAm), see Fig. 4a. Preliminary studies with PAAm only show that at an AAm concentration of least 0.3 M is necessary to generate a full stress percolation PAAm network (Supplementary Fig. 10a). At $t = 0\,s$, just after reaching the desired temperature, the storage modulus of the PIC gel decreases with increasing AAm concentration as a result of the Hofmeister effect[28]. In time, all composite hydrogels exhibit an increase in $G'$, which corresponds to the formation of the PAAm network, analogous to the samples without PIC present (Supplementary Fig. 10b). At the lowest AAm concentration, the formation of the PAAm network only leads to a very small increase in $G'$, whereas the modulus increases by more than an order of magnitude for the highest AAm concentrations. When we reverse the order in which the networks are formed by cooling and reheating these hybrid hydrogels (Supplementary Fig. 11), we observe that the modulus of the PIC/PAAm hybrid is not fully restored to its original value on reheating, which indicates that the fine mesh of the PAAm network hinders the reformation of the fibrous PIC network.

The PIC/PAAm composites all show a nonlinear stiffening response, even at the highest AAm concentrations where the linear mechanics are clearly dominated by the PAAm network (Fig. 4b). At low AAm concentrations, the hybrid hydrogels show an identical stiffening response as the pure PIC gels, with the stiffness increasing with stress as $K' \propto \sigma^{3/2}$. At the higher AAm concentrations, however, the hybrids show a delayed and slightly weaker stiffening response, which leads to a lower $K'$ in the high-stress regime. The stress necessary to deform the PAAm networks does not contribute to the strain-stiffening response of the PIC network and a denser PAAm network causes an increasing amount of stress directed to deform PAAm. In other words, increasing the AAm concentration increases the hydrogel stiffness in the low-stress linear regime, but decreases the stiffness of the gels in the nonlinear regime at high stress.

A different, but very common approach to tune the mechanics of PAAm hydrogels, or flexible polymer networks in general, is to change the crosslink density of the network[51]. We varied the amount of crosslinker MBAA from 0.25 to 5.0 mol% of AAm, while keeping the PIC and AAm concentrations constant at 1.0 and 36 mg ml$^{-1}$, respectively. In the linear regime, the stiffness of the hybrid hydrogels (Fig. 4c and Supplementary Fig. 12) increases for higher crosslink densities, but in the high-stress regime the opposite trend is observed, similar to the hydrogels with increasing AAm concentrations. At the highest MBAA

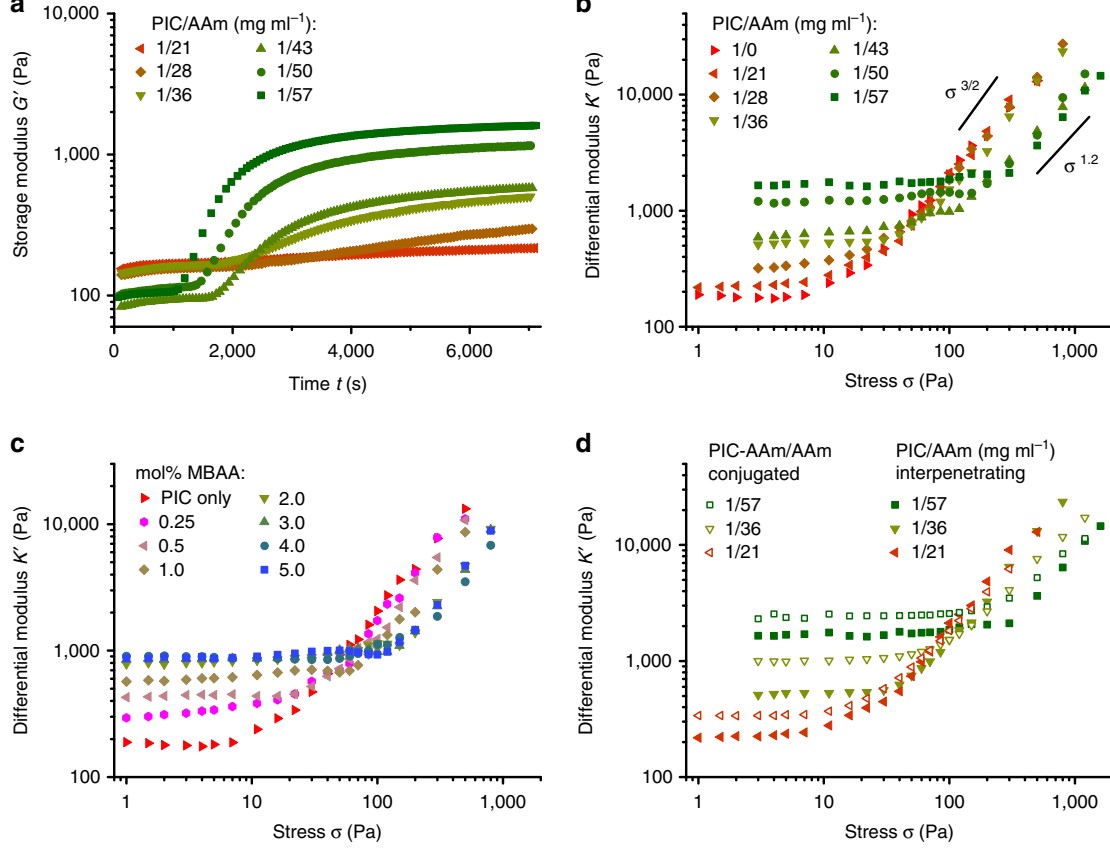

**Figure 4 | Mechanical properties of composites with semi-flexible PIC networks and flexible PAAm networks.** (**a**) Storage modulus $G'$ as a function of time for PIC/PAAm hybrid hydrogels at $T = 50\,°C$, showing the formation of the PAAm network within the pre-formed PIC network. De decrease in $G'$ with increasing (AAm) is the result of the Hofmeister effect. (**b**) The differential modulus $K'$ of the PIC/PAAm hybrid hydrogels increases with AAm concentration in the linear regime at low stress, but decreases with AAm concentration in the high-stress regime. The solid lines represent the slopes of the curves. (**c**) $K'$ as a function of stress for PIC/PAAm hybrid hydrogels at $T = 50\,°C$ with 36 mg ml$^{-1}$ (0.5 M) AAm and increasing concentrations of crosslinker MBAA. (**d**) $K'$ as a function of stress for conjugated (open symbols) and non-conjugated polymer networks (solid symbols, data from **b**).

concentrations, the effect diminishes and no further changes are observed in both the high- and low-stress regime.

Analogous to the PIC/fibrin hybrid hydrogels, we also conjugated the PAAm network to the PIC network through functionalization (2.0%) of the PIC polymers with AAm (PIC-AAm, see 'Methods' section, Supplementary Methods and Supplementary Fig. 13 for details), leading to covalent bonds between the two networks. These covalent interactions result in stiffer hydrogels compared to their interpenetrating counterparts (Fig. 4d and Supplementary Fig. 14), even for composites with low AAm content (purple triangles in Fig. 4d). Similar to the PIC/fibrin hydrogels, the bonds between the two networks increase the crosslink density of the composite network, resulting in an increased linear modulus. In the high-stress regime, the strain-stiffening response of the PIC network is hardly affected by the covalent interactions with the PAAm network and the mechanics are largely dominated by the PAAm concentration and crosslink density.

Whereas stiff rods increase the sensitivity of strain-stiffening hydrogels, the addition of a network of flexible polymers reduces the stiffening response of the resulting hydrogels. The flexible polymer network provides several approaches to increase the linear modulus of the composite gels, through increasing the polymer concentration, the crosslink density or by conjugating the two networks. For all approaches, however, a higher gel stiffness simultaneously decreases the nonlinear stiffening response of the double-network hydrogels, by making them less sensitive to an applied stress. One should also consider that the use of the flexible polymers in relatively high concentrations removes some of the architectural advantages of the semi-flexible polymer networks, such as its characteristically high porosity.

## Discussion

The persistence length and the network mesh size are the key length scales that control the linear and nonlinear mechanical properties of hydrogels. In hydrogels of semi-flexible filaments these length scales are of a similar order, which gives rise to a rich mechanical behaviour[15,26,52] that Nature uses on many occasions. In biological materials, a high persistence length is often realized through the bundling of individual filaments. The controlled formation of high persistence length bundles of synthetic polymers, however, has proven very difficult to achieve. The approach we present in this work controls the (non)linear mechanical response of polymer networks by combining polymers with differing persistence lengths into hybrid polymer networks. This introduces additional characteristic length scales in the network, which allows us to tailor the mechanics. For instance, stiff rod-like fibres (high persistence length and high contour length) that supress non-affine deformations of the network, promote the stiffening response of a semi-flexible hydrogel. On the other side of the spectrum, adding flexible polymers (low persistence length, low pore size) that do not stiffen under stress, reduce this stiffening response. The combination of multiple semi-flexible polymer networks in one material is especially interesting, because here, the (non)linear mechanical properties of the resulting material not only depend on the ratio between the components, but also on the nature of the interactions between them.

Nature uses a similar approach of combining stiff, semi-flexible and flexible fibres into biological composite networks. In the cytoskeleton for example, semi-flexible actin and intermediate filaments will all contribute to the strain-stiffening response of the network, and the stiff microtubules will further enhance the sensitivity to an applied stress. In the extracellular matrix, flexible elastin fibres may decrease the nonlinear response of semi-flexible fibrin and collagen networks. But how exactly these combinations

affect the mechanical properties of the resulting composite materials has barely been looked into, especially in terms of the nonlinear strain-stiffening properties[23,24].

Even though cells are able to stretch their matrix well into the nonlinear regime[53], the vast majority of regenerative medicine studies merely considers the simple linear modulus of the hydrogel matrix[3]. Our results indicate that the nonlinear mechanics of composite polymer networks are different from their single-component mimics, and we present an approach to systematically vary the nonlinear mechanics by varying the persistence length of the components, which should lead to a better understanding of the effect of the mechanics, including the nonlinear mechanics on cellular behaviour. Ultimately, the combination of polymers with different mechanical properties into hybrid hydrogels will help the design of a next generation of responsive materials and tuneable artificial extracellular matrices for tissue engineering applications.

## Methods

**Materials.** Stabilized CNT dispersions in water were purchased from Nanostructured & Amorphous Materials, Inc. (Houston, TX) and were used as received. Three different samples were studied: multi-walled CNTs (length $L \approx 10\,\mu m$, diameter $d \approx 20\,nm$), single-walled ($L \approx 10\,\mu m$, $d \approx 1.8\,nm$) and short single-walled CNTs ($L \approx 1\,\mu m$, $d \approx 1.8\,nm$). Tube lengths and diameters were given by the supplier, the latter were confirmed by transmission electron microscopy experiments (see Supplementary Fig. 2). Fibrinogen (from bovine plasma), AAm, MBAA and potassium persulfate were purchased from Sigma-Aldrich. Unfunctionalized[27,54] and azide-functionalized[32,33] PICs were synthesized following earlier reported procedures. The monomer to catalyst ratio for all polymers was 2,000:1, which resulted in polymers with molecular weights of $\sim 400\,kg\,mol^{-1}$, based on viscometry measurements. Different batches of polymer used in this study did not differ more than 10% in molecular weight. The degree of azide functionalization was for the peptide conjugation 0.2% and for the AAm conjugation 2.0%. These polymers, also polymerized with a monomer to catalyst ratio of 2,000:1, showed similar molecular weights. The synthesis of BCN-functionalized AKQAGDV peptides and DBCO-functionalized AAm is given in the Supplementary Methods.

**Sample preparation.** For PIC/CNT composites, PIC was dissolved in $18\,M\Omega\,cm$ purified water by stirring in a cold room at $4\,°C$ for at least 24 h. The PIC solution was mixed with a pre-cooled, freshly sonicated CNT dispersion in a 1:1 ratio on ice. The mixture was transferred to the rheometer and immediately heated to $37\,°C$. To form a PIC/fibrin composite, PIC ($2\,mg\,ml^{-1}$) was dissolved in DMEM-Hepes cell culture medium (Gibco) with 10% foetal bovine serum (Gibco) and pen/strep by stirring in a cold room at $4\,°C$ for at least 24 h. Fibrinogen was dissolved in PBS buffer and the PIC and fibrinogen solutions were mixed in a 1:1 ratio on ice. After transferring the mixture to the rheometer and immediate heating to $37\,°C$, the sample was incubated for 1 h. For the peptide-conjugated samples, the same procedure was used, only with the peptide-functionalized PIC (see Supplementary Methods). For the PIC/PAAm composite gel, a PIC solution in $18\,M\Omega\,cm$ purified water ($4\,mg\,ml^{-1}$, stirred for 24 h at $4\,°C$) was mixed on ice with AAm ($4\,M$) and MBAA ($0.1\,M$) solutions, both in $18\,M\Omega\,cm$ purified water. To this mixture, $K_2S_2O_8$ was added at a final concentration of 10 mM. The mixture was transferred to the rheometer and heated immediately to $50\,°C$ to initiate AAm polymerization. PIC/PAAm composite hydrogels were obtained after at least 2 h of incubation at $T = 50\,°C$. For the AAm-conjugated hybrids, an azide-functionalized PIC (2.0% azide functionalization) solution in $18\,M\Omega\,cm$ purified water (stirred for 24 h at $4\,°C$, $4\,mg\,ml^{-1}$, 1 ml) was mixed with DBCO-AAm (Supporting Material compound 5, $1.7\,mg\,ml^{-1}$ in DMSO, $26\,\mu l$) on ice. After 5 min, the PIC solution was mixed with solutions of AAm, MBAA and potassium persulfate, transferred to the rheometer and heated immediately to $50\,°C$.

**Rheology.** Rheological measurements were performed on a stress-controlled rheometer (Discovery HR-1 or HR-2, TA Instruments) using a aluminium or steel parallel plate geometry with a plate diameter of 40 mm and a gap of $500\,\mu m$. All samples were loaded into the rheometer in the liquid state at $T = 5\,°C$. PIC/CNT samples were heated at a rate of $1.0\,°C\,min^{-1}$ to $T = 37\,°C$, during which the complex modulus $G^\star$ was measured by applying an oscillatory deformation of amplitude $\gamma = 0.04$ at frequency $\omega = 1.0\,Hz$. To determine the linear modulus, the sample was equilibrated at $37\,°C$ for 10 min after which the storage modulus $G'$ was measured in oscillation with $\gamma = 0.04$ and $\omega = 1.0\,Hz$. PIC/fibrin samples were heated to $37\,°C$ immediately after loading the sample and the complex modulus $G^\star$ was measured in oscillation with $\gamma = 0.01$ and $\omega = 1.0\,Hz$ for one hour. PIC/PAAm samples were heated to $50\,°C$ immediately after loading the sample and the complex modulus $G^\star$ was measured in oscillation with $\gamma = 0.01$ and $\omega = 1.0\,Hz$ for at least 2 h, or until the storage modulus $G'$ reached a constant

value. Drying of the samples was prevented by maintaining a humid atmosphere and covering the edge of the sample with silicon oil. The nonlinear stiffening regime was studied at 37 °C for the PIC/CNT and PIC/fibrin hydrogels and at 50 °C for the PIC/PAAm hydrogels. A pre-stress protocol was used[40], where the gel was subjected to a constant pre-stress $\sigma = 0.1–1,000$ Pa with a small oscillatory stress superposed at a frequency of $\omega = 10–0.1$ Hz to determine the differential modulus $K'$. The oscillatory stress was at least 10 times smaller than the applied pre-stress and $K'$ was independent of frequency for all samples.

**Data availability.** All data of the work presented in this manuscript are available from the authors on reasonable request.

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

## Acknowledgements

We acknowledge financial support from NWO Gravitation (Grant 024.001.035) and NanoNextNL.

## Author contributions

M.J. and P.H.J.K. designed and interpreted the mechanical studies. M.J., S.L.V. P.v.S. and D.V. synthesized the materials and carried out mechanical tests. A.E.R. and P.H.J.K. supervised the project. All authors contributed to the manuscript.

## Additional information

**Competing interests:** The authors declare no competing financial interests.

**Publisher's note**: 

