## [Peer Review File · Nature Communications]

Reviewers' comments:

Reviewer #1 (Remarks to the Author):

This is a highly interesting and innovative study of the elastic response of composite networks formed by compatible polymers with different persistence lengths and connectivity. The experimental system is really great and the results open up new opportunities in materials science and soft matter physics. Overall, the results will have significant impact in hydrogel and biopolymer mechanics studies and the rheological studies are generally sound and convincing.

There are rheological data providing evidence that the different type of polymers are compatible and do not micro phase-separate, but more direct evidence is needed. Even simple scattering intensity or turbidity data of PIC gels with the different filaments would be very informative.

Another issue relates to interpretation of the effects of CNTs, which are presumed not to attach to the PIC networks but interact only sterically. Here theories by Doi or Odijk for example would seem important for interpreting these results and should be considered.

Also the data here report only elastic moduli. It would be good to know if there are also differences in G'' or in the G' or K' results at different frequencies.

Fig 1. It would be useful to add the chemical structure of PIC. What is its degree of polymerization?

Calling carbon nanotubes stiff as opposed to fibrin bundles as soft is a bit ambiguous. The persistence length of a single walled carbon nanotube is on the order of 10 microns, so not much different from fibrin fibers. Only the multiwalled ones have persistence lengths of mm. Presumably these are all surfactant stabilized? How does the nature of the surfactant affect gelation? More detail is needed in the main text to define "polymer-stabilized CNTs".

It would also seem important to know what the contour length of a typical fibrin filament is or the mesh size of the network. Larger or smaller than its persistence length?

p.4 at the various concentrations, are the three types of CNTs above or below their overlap concentration, or c^* . (i.e. is $n \times l^3$ more or less than the sample volume with n = number of filaments and l their length) This criterion would seem essential for interpreting their effect on K' . Presumably they are always below the Onsager transition for nematic phase?

p.5. Here again considering the length of the multiwalled CNTs is critical. If they are above the overlap concentration, then their effect is not really astounding. It just reflects the shift in rheology that occurs above the overlap concentration.

There should also be some discussion about dissipation caused by the non-attached rods and filaments. At moderate to high strains they should produce a large viscous relaxation in the time scale of the measurements. Is this seen? What does stress relaxation of the networks +/- CNT look like?

Why was potassium persulfate used instead of the more common ammonium persulfate?

Reviewer #2 (Remarks to the Author):

This is an interesting manuscript discussing how the rheological properties of a PIC gel can be affected by the addition of various additives and networks. Overall, the goals and aims are clear, the work is generally to a high standard and I would think that this manuscript will be of interest to a significant community. As such, I support publication after the following have been addressed.

It would be useful to include some example rheological data in the main text, as opposed to relying only on the interpreted data in Fig. 2. This would allow a more direct, immediate comparison to the data in Fig. 3 for example.

It would also be useful is some description of the mechanical properties of each of the CNT samples were provided. Are the persistence lengths significantly different for example? Also, it is not clear how the lengths are assigned from the data provided in Fig. S1. More data needs to be provided here.

The two-component data in Fig. 3 is very nice! The data are very convincing and it is very satisfying to see such a perfect temperature sweep for example.

Is the PIC network affected at all by the presence of the free radical initiator? The G' at the start of the experiment in Fig. 4a seems quite binary; around 22 Pa up until a ratio of 1/36, and then around 80-100 Pa for higher ratios. Why is this?

For all of the rheology data, it would be interesting and useful to see the effect on G'' as well as G' .

Reviewer #3 (Remarks to the Author):

This is yet another interesting paper from these authors on their biomimetic polymer gels based on the PIC polymer. This time around they explore hybrid (double) gel network of PIC with stiff, semi-stiff and flexible components, respectively. The results show a trend which they relate back to what has been observed in biology. As far as this reviewer can tell, the data and the results are of good quality.

As in their previous work, they come to the conclusion that this is all about what they call the nonlinear mechanism of the gel, based on measuring K' as a function of stress. This reviewer will admit having difficulty in understanding what K' vs stress really means. Hence:

i) Could the authors please add some conventional rheology data, i.e., strain and frequency sweeps for G' and G'' ? The former is particularly important to reveal whether and how the linear viscoelastic region (LVE) changes in these hybrid networks. This reviewer appreciates that if the authors did not collect this data at the time of performing all the other experiments, asking for this information for all the different systems and mixtures described here would be extremely time consuming. However, the authors ought to be able to do these measurement on a selected collection of their hybrid systems plus of course PIC itself. Obviously, if the authors do see any interesting changes in the LVE, they need to discuss these. Perhaps there is no connection between LVE and K' vs stress but for those of us more familiar with 'linear' rheology this would still be useful and help us to appreciate what K' vs stress is.

ii) Secondly, with regards to changes in the microstructure: Did the authors perform any characterization of these hybrid gels using microscopy methods? If so, it would be interesting to add these (even if only some representative examples). This would enforce some of the claims made here.

iii) Finally, and again, this seems also absent or unclear from earlier papers from these authors – could the authors perhaps explain why they think PIC has these mechanical properties? What factors are most important? Length? Aspect ratio? Molecular weight? Intramolecular interactions? It would be useful, e.g. to give others a clue to how to design similar or even better biomimetic gels. If PIC is really unique in this regard then that does diminish somewhat the utility of this work but if the design principles behind PIC can be applied more broadly, this will greatly increase the impact of this work.

In conclusion – the authors are probably right that no-one has attempted to manufacture hybrid gel systems with these nonlinear mechanical properties and in terms of biology, such gels are very important, deserving publication in Nature Comm. if they can address the point above.

Reply to reviewers' comments

Reviewer #1 (Remarks to the Author):

This is a highly interesting and innovative study of the elastic response of composite networks formed by compatible polymers with different persistence lengths and connectivity. The experimental system is really great and the results open up new opportunities in materials science and soft matter physics. Overall, the results will have significant impact in hydrogel and biopolymer mechanics studies and the rheological studies are generally sound and convincing.

1. There are rheological data providing evidence that the different type of polymers are compatible and do not micro phase-separate, but more direct evidence is needed. Even simple scattering intensity or turbidity data of PIC gels with the different filaments would be very informative.

Reply: The reviewer is right; this information is very useful. Unfortunately, the characteristic length scales of the PIC polymer are 3 nm (bundle size) and 100 nm (pore size), which make optical techniques and turbidity measurements useless. All PIC gels and the PIC/CNT and PIC/PAcrAm composites are fully transparent. We are working on super-resolution, with limited success so far. Fibrin has a larger pore size, which can be visualised with (fluorescence) microscopy (Ref: Bruekers et al. Cell Adhesion Migration 2016, # 45 manuscript). We also did do several SAXS experiments where we reliably find the gel architecture. In the composites, however, we simply find the sum of the two components. In the manuscript, we addressed the compatibility between the components in the Materials and Methods section of the R&D.

2. Another issue relates to interpretation of the effects of CNTs, which are presumed not to attach to the PIC networks but interact only sterically. Here theories by Doi or Odijk for example would seem important for interpreting these results and should be considered.

Reply: See remark 7 of reviewer 1, where a similar point is raised.

3. Also the data here report only elastic moduli. It would be good to know if there are also differences in G'' or in the G' or K' results at different frequencies.

Reply: By default, we measure the mechanical properties as a function of frequency, also in the nonlinear regime (where we do frequency sweeps on a pre-stressed sample). In all our measurements, the frequency behaviour is dominated by the PIC gel, which has a very strong elastic component and the loss is negligible in the frequency regime that we study experimentally.

Reviewer 1, and also reviewers 2 and 3 make us realise that this is not obvious to the reader and consequently, we inserted some experimental data in the manuscript and in the supporting information. We provide zero stress frequency sweeps of G' and G'' as well as the high stress K' and K'' (Figure S4). All the data indeed show that there is no effect in time, not even at high stress or at "high" CNT loading. Besides adding the data to Figure 2 and the SI, we addressed this issue in the Results section.

4. Fig 1. It would be useful to add the chemical structure of PIC. What is its degree of polymerization?

Reply: This is a good suggestion. We modified Figure 1 to include the chemical structure of the polymer as well as the sketch. The molecular weight ($M_v = 400$ kg/mol), as well as the degree of polymerisation and the contour length were added to the figure.

5. Calling carbon nanotubes stiff as opposed to fibrin bundles as soft is a bit ambiguous. The persistence length of a single walled carbon nanotube is on the order of 10 microns, so not much

different from fibrin fibers. Only the multiwalled ones have persistence lengths of mm. Presumably these are all surfactant stabilized? How does the nature of the surfactant affect gelation? More detail is needed in the main text to define “polymer-stabilized CNTs”.

Reply: *The reviewer is right and we used the term rigid a bit too liberal for the SW-CNTs, which are only an order of magnitude stiffer than the fibrin fibers that we measured. This was adapted in the manuscript. Indeed, the CNTs are stabilised by a (polymer) surfactant. We see no effects on the gelation process of the CNTs with the surfactant (Figure S3). This was further highlighted in the manuscript.*

6. It would also seem important to know what the contour length of a typical fibrin filament is or the mesh size of the network. Larger or smaller than its persistence length?

Reply: *Fibrin forms a continuous branched network, so the contour length is poorly defined. The mesh size is typically between 1-2 μm (dependent on the preparation of the sample), which is a bit smaller than the persistence length of about 10 μm . We mentioned these dimensions in the manuscript (page 3).*

7. p.4 at the various concentrations, are the three types of CNTs above or below their overlap concentration, or c^* . (i.e. is $n \times l^3$ more or less than the sample volume with n = number of filaments and l their length) This criterion would seem essential for interpreting their effect on K' . Presumably they are always below the Onsager transition for nematic phase?

p.5. Here again considering the length of the multiwalled cnts is critical. If they are above the overlap concentration, then their effect is not really astounding. It just reflects the shift in rheology that occurs above the overlap concentration.

Reply: *This remark comes back to remark 2 of the reviewer and addresses in which concentration ranges the CNTs are added and what interactions could be expected. Even at the highest concentrations that we added, all CNT samples are still in the semi-dilute regime, below c^* and certainly below the Onsager transition. This means that steric interactions are present, but not dominating the mechanical interactions. To address the questions of the reviewer, we experimentally measured the mechanical properties of the CNT solutions (at the highest concentrations used) as a function of strain rate (and added them to the Supplementary Material, Figure S2). Only at very high strain rates, much higher than we have in our experiments, we see the effect of the CNTs by an increase in the viscosity. We conclude that in the PIC gels, we really see the effect of increasing the affinity of the network. As other readers may have similar questions, we addressed this briefly in the manuscript.*

8. There should also be some discussion about dissipation caused by the non-attached rods and filaments. At moderate to high strains they should produce a large viscous relaxation in the time scale of the measurements. Is this seen? What does stress relaxation of the networks +/- CNT look like?

Reply: *Similar to the previous point, the remark of the reviewer prompted us to carefully review our experimental data. We do not see the effects mentioned by the reviewer. Even at the highest CNT concentrations and the highest strain, the frequency sweeps do not indicate dissipation of energy, that is, not at the time scale of interest. The plots are given in the manuscript for the CNT (Figure 2a) and for the other materials in the Supplementary Material (Figure S4).*

9. Why was potassium persulfate used instead of the more common ammonium persulfate?

Reply: *Potassium and ammonium persulfate can both be used as thermal initiator. The former (that we had on the shelf) has a lower solubility in water, but at the concentrations that we used it, this is no problem whatsoever.*

Reviewer #2 (Remarks to the Author):

This is an interesting manuscript discussing how the rheological properties of a PIC gel can be affected by the addition of various additives and networks. Overall, the goals and aims are clear, the work is generally to a high standard and I would think that this manuscript will be of interest to a significant community. As such, I support publication after the following have been addressed.

1. It would be useful to include some example rheological data in the main text, as opposed to relying only on the interpreted data in Fig. 2. This would allow a more direct, immediate comparison to the data in Fig. 3 for example.

Reply: This is a good remark that allows for comparison between different cases. We already provided some of the data requested in the Supplementary Material. Based on the comment of the reviewer, we transferred it to the manuscript (Figure 2b) and discussed it in the main text.

2. It would also be useful is some description of the mechanical properties of each of the CNT samples were provided. Are the persistence lengths significantly different for example? Also, it is not clear how the lengths are assigned from the data provided in Fig. S1. More data needs to be provided here.

Reply: The persistence length for SW and MW-CNTs were not measured experimentally, but retrieved from the literature. Also, we did not experimentally determine the length, but used the lengths provided by the supplier. The tube diameters (also given by the supplier) were confirmed by the TEM experiments in Figure S1. We updated the manuscript with the requested data.

3. The two-component data in Fig. 3 is very nice! The data are very convincing and it is very satisfying to see such a perfect temperature sweep for example.

Reply: We agree with the reviewers and share the enthusiasm.

4. Is the PIC network affected at all by the presence of the free radical initiator? The G' at the start of the experiment in Fig. 4a seems quite binary; around 22 Pa up until a ratio of 1/36, and then around 80-100 Pa for higher ratios. Why is this?

Reply: This is well remarked by the reviewer and we should have commented on the differences in G_0 in Figure 4a. In fact, these differences are not caused by the initiator (which in all samples has the same concentration), but due to the presence of the acrylamide monomer. Addition of the relatively hydrophobic monomer (compared to water) increases the gel temperature following the Hofmeister effect, which for the PIC gels means a small decrease in G_0 (Jaspers et al. Adv. Funct. Mater. 2015). We explained the effect in the manuscript.

5. For all of the rheology data, it would be interesting and useful to see the effect on G'' as well as G' .

Reply: We provided G' , G'' as well as K' and K'' for all different samples in the Supplementary Material (Figure S4). For all composites, the elastic component G' and K' dominate the viscous components G'' and K'' and frequency data is rather featureless. As an example, we included the data for the PIC/CNT hybrids in the main text and Figure 2a.

Reviewer #3 (Remarks to the Author):

This is yet another interesting paper from these authors on their biomimetic polymer gels based on the PIC polymer. This time around they explore hybrid (double) gel network of PIC with stiff, semi-stiff and flexible components, respectively. The results show a trend which they relate back to what has been observed in biology. As far as this reviewer can tell, the data and the results are of good quality.

As in their previous work, they come to the conclusion that this is all about what they call the nonlinear mechanism of the gel, based on measuring K' as a function of stress. This reviewer will admit having difficulty in understanding what K' vs stress really means. Hence:

1. Could the authors please add some conventional rheology data, i.e., strain and frequency sweeps for G' and G'' ? The former is particularly important to reveal whether and how the linear viscoelastic region (LVE) changes in these hybrid networks. This reviewer appreciates that if the authors did not collect this data at the time of performing all the other experiments, asking for this information for all the different systems and mixtures described here would be extremely time consuming. However, the authors ought to be able to do these measurement on a selected collection of their hybrid systems plus of course PIC itself. Obviously, if the authors do see any interesting changes in the LVE, they need to discuss these. Perhaps there is no connection between LVE and K' vs stress but for those of us more familiar with 'linear' rheology this would still be useful and help us to appreciate what K' vs stress is.

Reply: The reviewer is quite right: we sometimes forget that a significant fraction of the soft matter community may not be fully acquainted with nonlinear mechanical properties. Of course, we always measure the LVE. We prefer to use the pre-stress protocol (that yields $K' = \partial\sigma/\partial\gamma$) over a 'standard' LAOS experiment (yielding $G' = \sigma/\gamma$) or strain sweep as the first is a more gentle method and also provides frequency information. In addition, we find that K' does more accurately represent the mechanical properties of the material at a specific stress (or strain). To address the question of the reviewer (and most probably a large part of the readership), we added some 'standard' curves to the manuscript (Figure 2a) and added a scheme that clearly explains the difference between different moduli to the Supplementary Material, Scheme S3.

2. Secondly, with regards to changes in the microstructure: Did the authors perform any characterization of these hybrid gels using microscopy methods? If so, it would be interesting to add these (even if only some representative examples). This would enforce some of the claims made here.

Reply: The reviewer makes an excellent suggestion to further study the microstructure of the gels. Unfortunately, the characteristic length scale (pore size) of the PIC gels is about 100 nm, depending on the concentration, which makes it just not suitable for optical or fluorescent microscopies. Our preliminary efforts to super-resolution microscopy have not yielded success yet, but we are still trying. Particularly, we look forward to study the network changes at high stress. We hope to show this in the future (see also comment 1 of Reviewer 1). As mentioned before, the fibrin samples were studied by confocal fluorescence microscopy before (Ref: Bruekers et al. Cell Adhesion Migration 2016), which is mentioned in the manuscript.

3. Finally, and again, this seems also absent or unclear from earlier papers from these authors – could the authors perhaps explain why they think PIC has these mechanical properties? What factors are most important? Length? Aspect ratio? Molecular weight? Intramolecular interactions? It would be useful, e.g. to give others a clue to how to design similar or even better biomimetic gels. If PIC is really unique in this regard then that does diminish somewhat the utility of this work but if the design principles behind PIC can applied more broadly, this will greatly increase the impact of this work.

Reply: The reviewer asks for design principles for semi-flexible, strain-stiffening networks to increase the impact of the manuscript. This is not an easy question to answer, although we made an attempt in our Nat. Commun. 2014 and Biomacromolecules 2016 papers. In short, we find these intriguing mechanical properties when the mesh size in the network is similar or slightly smaller than the persistence length of the network fibres. For (synthetic) polymers, the polymer-chain persistence length is not large enough to realise this and these numbers can only be achieved by bundling; in this case, the polymer length is not very important any longer, as long as

it is large enough to form proper bundles. We started the Discussion section with these conditions.

4. In conclusion – the authors are probably right that no-one has attempted to manufacture hybrid gel systems with these nonlinear mechanical properties and in terms of biology, such gels are very important, deserving publication in Nature Comm. if they can address the point above.

***Reply:** We hope that we have satisfactorily addressed the comments and questions of reviewer 3 as well as reviewers 1 and 2.*

REVIEWERS' COMMENTS:

Reviewer #1 (Remarks to the Author):

The authors have added substantial new material to the text to address the previous issues raised. I think this revised version is significantly improved and requires no additional changes.

Reviewer #2 (Remarks to the Author):

The authors have addressed my comments and I am happy for the paper to be published in its current form.

Reviewer #3 (Remarks to the Author):

The authors should be thanked for addressing most of the comments from the reviewers so thoroughly. What was already an interesting paper is now even better.

The authors have taken on board the point about the difference between 'linear' and 'non-linear' rheology. This is to be applauded. I was a little disappointed or confused with the authors reply to my question about conventional rheology plots. The authors have now included several frequency sweeps for their gels. However, the acid test for the linear viscoelastic region (LVE) is a strain sweep, not frequency. The 5% rule in a strain sweep is sometimes used to characterise the LVE. Additionally, the cross-over point in the strain sweep is often of interest. As with my previous comments, I don't expect the authors to provide a strain sweep for all their systems but at least for PIC itself. It would also be extremely useful if they updated Figure S3 with this in mind, i.e. to show also the difference (if any) in the strain-sweep behavior of 'linear' and 'non-linear' polymers. One thing I didn't get with Figure S3, why are the y-axis not the same on the two right-hand panels?

One of the other reviewers also raised the point about microstructure characterisation. I appreciate that attempts to use light microscopy or scattering methods have not work but what about TEM, especially cryogenic TEM? Or even freeze-fracture SEM? If this can't be done or returns no useful information so be it.

With regards to the question about design principles, I was aware of the authors' idea that it was all about mesh size and persistence length. My question was really directed at the level below that – how does one control the mesh size or persistence length by molecular design? I am happy to leave the authors answer as it is but if they wish to expand on their answer that would be welcomed.

REVIEWERS' COMMENTS:

We thank both reviewers 1 and 2 for their positive evaluation and their proposal to publish the manuscript. The additional remarks of reviewer 3 are addressed below.

Reviewer #1 (Remarks to the Author):

The authors have added substantial new material to the text to address the previous issues raised. I think this revised version is significantly improved and requires no additional changes.

Reviewer #2 (Remarks to the Author):

The authors have addressed my comments and I am happy for the paper to be published in its current form.

Reviewer #3 (Remarks to the Author):

The authors should be thanked for addressing most of the comments from the reviewers so thoroughly. What was already an interesting paper is now even better.

The authors have taken on board the point about the difference between 'linear' and 'non-linear' rheology. This is to be applauded. I was a little disappointed or confused with the authors reply to my question about conventional rheology plots. The authors have now included several frequency sweeps for their gels. However, the acid test for the linear viscoelastic region (LVE) is a strain sweep, not frequency. The 5% rule in a strain sweep is sometimes used to characterise the LVE. Additionally, the cross-over point in the strain sweep is often of interest. As with my previous comments, I don't expect the authors to provide a strain sweep for all their systems but at least for PIC itself. It would also be extremely useful if they updated Figure S3 with this in mind, i.e. to show also the difference (if any) in the strain-sweep behavior of 'linear' and 'non-linear' polymers.

Reply: We understand the confusion of the reviewer. We obtain equivalent results to the experiments he/she suggests, but in a different experiment. Since the readership may experience the same challenges, we addressed it again, hopefully better than in the previous revision. In short, rather than ramping up the strain and measuring the stress in the rheometer (a stress-controlled rheometer actually does the opposite), we now apply an increasing constant stress and measure the mechanical properties with an additional small oscillatory deformation. Actually, a very similar experiment, to which the polymer gel should respond the same. A comparison between the two methods is given by Broedersz et al in 2010 (reference 39 in the manuscript).

To compare theory and reality, we did a simple strain ramp on the PIC gel and compared the results to our default pre-stress method on the same sample. The results are in the Supplementary Figure 1c-f. One sees that the data from both methods overlap (panel f). At high stress, the pre-stress method is a bit more accurate, since the oscillatory deformations are small. We hope that these experiments are sufficiently clear to convince the reviewer and the readers.

One thing I didn't get with Figure S3, why are the y-axis not the same on the two right-hand panels?

Reply: The difference between the two graphs is the concentration PIC. A higher concentration yields a stiffer gel (with higher G' .) Hence, the adaption to the y-axis scale.

One of the other reviewers also raised the point about microstructure characterisation. I appreciate that attempts to use light microscopy or scattering methods have not worked but what about TEM, especially cryogenic TEM? Or even freeze-fracture SEM? If this can't be done or returns no useful information so be it.

Reply: Again, the reviewer is right and any EM specialists that we speak to cannot believe the problems we face with obtaining reliable micrographs that do not show significant sample preparation artefacts. It is work in progress at the moment.

With regards to the question about design principles, I was aware of the authors' idea that it was all about mesh size and persistence length. My question was really directed at the level below that – how does one control the mesh size or persistence length by molecular design? I am happy to leave the authors answer as it is but if they wish to expand on their answer that would be welcomed.

Reply: This is a very interesting question that the reviewer poses: How do we make stiff polymers? It is not so easy to answer (and we have not in the manuscript), but in our opinion, a twisted, helical organisation of the backbone is a key design parameter.